# Effect of Triweekly Interdental Brushing on Bleeding Reduction in Adults: A Six-Month Retrospective Study

**DOI:** 10.3390/healthcare9091239

**Published:** 2021-09-21

**Authors:** Jae-Young Lee, Seon-Jip Kim, Hyo-Jin Lee, Hyun-Jae Cho

**Affiliations:** 1Department of Dental Hygiene, College of Health Science, Dankook University, Cheonan 31116, Korea; dentaljy@dankook.ac.kr; 2Department of Preventive Dentistry and Public Oral Health, School of Dentistry, Seoul National University, Seoul 03080, Korea; 1177155@snu.ac.kr; 3Dental Research Institute, Seoul National University, Seoul 03080, Korea; 4Department of Dental Hygiene, College of Dentistry and Research Institute of Oral Science, Gangneung-Wonju National University, Gangeung 25457, Korea

**Keywords:** gingival bleeding, gingivitis, index, interdental brush, oral health behavior

## Abstract

This study assessed the effect of triweekly interdental brushing for a period of 6 months using the bleeding on full-mouth interdental brushing (BOFIB) index. All participants answered questionnaires and were instructed to clean all interdental areas using an interdental brush at least every second day. A dentist assessed the BOFIB index of 28 participants at baseline and at 3- and 6-month follow-up visits. We stratified the participants into three groups: IB-NN, those who used the interdental brush less than three times per week both before and after this program; IB-NY, those who used the interdental brush less than three times per week before the program but at least three times per week after the program; and IB-YY, those who used it at least three times per week both before and after the program. Owing to the weekly number of interdental brushings, the IB-YY and the other two groups showed a significant difference at baseline. At the 6-month follow-up, the weekly number of interdental brushings led to a significant difference between the IB-NN and the other two groups. The BOFIB index was lower among the compliant participants than among the noncompliant participants after 3 and 6 months of triweekly interdental brushing.

## 1. Introduction

Toothbrushing is an important method for controlling dental biofilms, an etiologic factor for periodontitis [1]. Lang et al. suggested that patients who practiced effective oral hygiene procedures at 48 h intervals did not develop gingivitis [2]. A systematic review suggested that toothbrushing removes only 42% of the entire dental biofilm [3]. Most oral diseases, such as periodontitis and dental caries, begin in the interproximal areas [4,5]. Hence, interdental cleaning at least every alternate day is crucial to prevent gingival inflammation. 

A meta-analysis suggested that interdental brushing was the most effective method for cleaning residual plaque in interproximal areas [6]. Although young adults were periodontally healthy, interdental hygiene requirements in this population were high, and most of the interproximal areas could be accessed by an interdental brush [7]. Waerhaug [8] suggested that interdental brushing could remove plaque from 2.0–2.5 mm below the gingival margin. However, longitudinal clinical investigations have not supported the effects of using interdental brushes. Furthermore, there is little evidence for oral professionals to refer to while educating dental patients on the method and frequency of interdental brushing to manage healthy gingival tissues.

An interdental brush can be used as a tool for periodontal assessment. Hofer et al. [9] reported that interdental brushes could be a valid alternative to a periodontal probe for assessing marginal bleeding in patients with gingivitis. Rosenauer et al. [10] also suggested that the interdental brush could detect potential pathological processes in the interproximal areas; they also demonstrated a significant decrease in bleeding tendency in the group that used an interdental brush, relative to the control group. Bourgeois et al. [11] also investigated the use of an interdental brush as a tool to measure periodontitis and showed that daily use of calibrated interdental brushes reduced interdental bleeding.

Studies have investigated the use of interdental brushes to assess gingivitis and gingival health using indices such as the Eastman interdental bleeding (EIB) [12], bleeding on interdental brushing (BOIB) [9], and bleeding on brushing (BOB) index [10]. These indices mainly consider gingival bleeding at the interdental sites because one of the characteristics of gingivitis is bleeding upon provocation [13]. Biofilm-induced gingivitis is characterized by biofilm formation at the gingival margin, changes in gingival color and contour, changes in sulcular temperature, increased gingival exudation, absence of attachment loss and bone loss, and histological changes. The ideal parameter should be objective (i.e., not susceptible to subjective interpretation), inexpensive, not time consuming, and easy to use for clinicians to ensure widespread application [14]. Thus, gingival health, including the presence of dental plaque and gingival bleeding, needs to be assessed quickly with minimal instrumentation.

Patients are often noncompliant with interdental cleaning because they render it unimportant or the interdental plaque, which is not visible, harmful. Moreover, the time and technique required to perform interdental cleaning also impedes compliance [15]. Thus, the degree of interdental irrigation and associated bleeding can be used as a measure of gum health, and through exponential assessment, utilizing the most commonly available interdental brushes. We hypothesized that it is possible to evaluate oral health. The purpose of this study is to identify the bleeding reduction effect of the triweekly interdental toothbrush using the bleeding on full-mouth interdental brushing (BOFIB) index over a period of 6 months. In this way, we tried to establish an objective basis for utilization for self-oral health management and specialist oral health management as one auxiliary index to assist the evaluation of gum health.

## 2. Materials and Methods

### 2.1. Study Design

This oral health program for promoting interdental brush use was conducted at a health center between March and November 2016 at Sungkyunkwan University, Seoul, Korea. This health center-based retrospective study was designed to compare BOFIB index scores in terms of interdental brushing habits. The study complied with the principles of the Declaration of Helsinki. Ethical clearance for the study was approved by the Institutional Review Board of the School of Dentistry, Seoul National University, Seoul, Korea (IRB number S-D2017008). 

### 2.2. BOFIB

The BOFIB was determined using an interdental brush (SKYDENT, 003 interdental brush for family, Korea) in the order as below.

#### 2.2.1. Selection of an Interdental Brush Size

Based on Table 1, a dentist (H-J) with extensive experience in the use of an interdental toothbrush visually examined the interdental space of the subjects and determined the size of the interdental brush (from SSS to M Table 1) that was able to pass through almost all of the interdental areas of the subjects (>90%). The size of the toothbrush was decided only at the baseline. At three months and six months of follow-up observation, the size of the interdental brush determined by baseline was still used.

#### 2.2.2. Decision of the BOFIB Index

The interdental brush was inserted horizontally on both the buccal and palatal or lingual sides in all accessible interdental areas. When moving from one interdental space to another, the dentist did not remove or wash dental plaque or blood of the interdental brush. The dentist then observed the bloodstained interdental brush and recorded the BOFIB index (Table 2) as follows: BOFIB0, no bleeding and only little white plaque noted after full-mouth interdental brushing; BOFIB1, no bleeding and moderate yellowish plaque noted after full-mouth interdental brushing; BOFIB2, slight bleeding after full-mouth interdental brushing; BOFIB3, abundant bleeding after full-mouth interdental brushing; and BOFIB4, abundant bleeding during full-mouth interdental brushing.

### 2.3. Procedure

The appropriate sample size for the study was calculated using a power and sample size calculation program (G Power 3.1.9.7) to estimate a sample size sufficient to detect differences in the BOFIB index between the groups based on interdental brushing compliance. The type I error probability (α) was 0.05, and the power (1-β) was 0.8. We initially estimated a sample of 26 participants but decided on a total sample size of 75 participants to account for possible loss to follow-up. A dentist performed all dental examinations. Seventy-six participants enrolled in this program; all participants had 28 teeth, excluding wisdom teeth. At the first visit, all participants were given a leaflet that described how to use an interdental brush and the interdental brushing technique using pictures and text. A dentist explained the use of an interdental brush and the interdental brushing technique to each participant, measured the BOFIB index, and recommended the primary interdental brush size. 

All participants were instructed to clean all interdental areas using an interdental brush at least every second day. After 3 months, only 54 participants revisited the program; the remaining 22 participants did not visit this center for personal reasons. At the 3-month follow-up visit, the participants were educated again about the correct use of interdental brushing, and BOFIB was measured using the same size interdental brush as in the first examination. After 6 months, 28 subjects participated in the final examination, and only 26 of them consented to this retrospective study and provided information about their dental records (Figure 1). 

### 2.4. Questionnaire

At baseline, the participants completed a questionnaire to provide information on demographics such as age; gender; weekly frequency of interdental brushing and interdental flossing; smoking (no/past/current); and health conditions, such as diabetes mellitus (no/yes). The weekly number of interdental brushing was recorded during each visit.

### 2.5. Study Groups

Based on compliance with the triweekly interdental brushing regimen before and after this program, we stratified the study participants into the following three groups: IB-NN (*n* = 17), those who used the interdental brush less than three times per week both before and after the program; IB-NY (*n* = 5), those who used the interdental brush less than three times per week before the program, but used it at least three times per week after the program; and IB-YY group (*n* = 4), those who used the interdental brush at least three times per week both before and after the program.

### 2.6. Statistical Analysis

All statistical analyses were performed using SPSS software (version 23.0; SPSS, Chicago, IL, USA). Statistical significance for all tests was set at α = 0.05. The differences between the IB-NN, IB-NY, and IB-YY groups for each variable were analyzed using chi-square tests for categorical variables and the Kruskal–Wallis H test for continuous variables. Bonferroni’s correction for multiple comparisons using the Mann–Whitney U test was performed (0.05/3 = 0.017) to analyze the nonparametric Kruskal–Wallis test results. The BOFIB index was analyzed by group and visit time using repeated measures analysis of variance (ANOVA) and Scheffe’s post hoc analysis.

## 3. Results

Table 3 shows the subject characteristics according to changes in oral health behaviors after interdental brushing education. The univariate associations between the change in habits for interdental brushing and explanatory variables were confirmed using a chi-square test and ANOVA. The mean age of the participants was 36.41 years, 51.80 years, and 52.00 years in the IB-NN, IB-NY, and IB-YY groups, respectively. The IB-YY group did not have past and current smokers. “S” was the largest interdental brush size used in this study. There were no significant differences between the groups in terms of gender, diabetes, smoking, or the number of weekly interdental flossing. At baseline, the weekly number of interdental brushing was significantly different between the IB-YY and the other two groups. However, at 6-month follow-up, the weekly number of interdental brushings was significantly different between the IB-NN and the other two groups.

Table 4 shows the change in the BOFIB index between the experimental groups after interdental brushing education (after 3 and 6 months). At baseline, the BOFIB scores were 2.88, 3.00, and 1.75 in the IB-NN, IB-NY, and IB-YY groups, respectively. The BOFIB score decreased after triweekly interdental brushing in all the groups, and the differences were significant (*p* < 0.001). In particular, the BOFIB score in the IB-NY group reduced to 1.40. The participants in the IB-NN group did not change their interdental brushing habits.

The main reasons for noncompliance included feeling uncomfortable or not being used to interdental brushing (64.7%), found it unnecessary (29.4%), experienced pain (11.8%), did not purchase the interdental brush (11.8%), and concerned about developing black triangles (loss of the interdental papillae) because of using the interdental brush (5.9%; Table 5).

## 4. Discussion

This study aimed to evaluate the reduction in gingival bleeding by triweekly use of an interdental brush for 6 months using full-mouth interdental brushing.

All participants were educated about the interdental brushing method and instructed to clean their interdental areas at least every second day. After 6 months, the BOFIB index reduced to 1.00 and 1.40 in the groups that used the interdental brush more than three times per week, and to 2.44 in the group that used it less than three times per week. Our results are consistent with those reported by previous studies [11,16] in which the daily use of an interdental brush showed a decrease in gingival bleeding. Bourgeois et al. evaluated interdental brushing in 46 adults without periodontal disease and reported that daily interdental brushing, there was reduced interproximal bleeding by 46% after one week, and by 72% after 3 months [11]. Jackson et al. compared the effects of an integral component of home plaque control in 77 chronic periodontitis cases [16] and demonstrated that the changes in plaque, papillae level, probing depth, and bleeding in the interdental brush group (*n* = 39) were greater than those in the floss group (*n* = 38) after six and 12 weeks.

The interdental brush can also be used to remove plaque in the gingival sulcus (up to a depth of 2.5 mm) that causes periodontal inflammation [8]. Interdental brushes allow the bristles to depress the interdental papilla and reach the subgingival areas. Thus, it can help decrease gingival inflammation. Dental floss and an interdental brush are commonly used to manage the interdental area. It may be easier to use an interdental brush when considering the method and time of use [15]. However, an interdental brush should be applied at sites of appropriate size, and dental floss can be used in narrow interdental spaces [17].

In this study, the subjects were instructed to use an interdental brush at least every second day. At the 6-month follow-up, we observed improvements in bleeding according to the participant’s compliance and management habits. In the participants who did not use an interdental brush at baseline but were compliant and usually performed triweekly interdental brushing, the amount of interdental bleeding decreased after 6 months. On the contrary, the interdental bleeding persisted in those who did not use an interdental brush at baseline and were noncompliant, including after educating them about interdental brushes.

The effect of oral health education varies according to the patient’s oral health status, habits, and compliance [15]. Therefore, oral health professionals need to identify the fac-tors that can increase patient compliance in oral health education and motivate them. Both internal and external factors can affect patient compliance [15]. The internal factors include fear and anxiety about visiting the dentist or hygienist, fear of pain or needles, lack of understanding, poor communication, apathy, perceived or actual lack of time, lifestyle, age, health beliefs, perceived unimportance of treatment or oral care, physical and psychological health, low self-esteem, and embarrassment. External factors include poor communication or involvement, stress, community influences, and socioeconomic status. Thus, oral health professionals should be aware of these factors and use efficient oral health education for their patients. In this study, the reasons reported by the study participants for not complying with the directions to consistently use an interdental brush were that they found it uncomfortable, non-habitual, unnecessary, or painful. Several participants did not purchase an interdental brush, and there were concerns about developing black triangles. Therefore, education on interdental brush use should include a proper explanation of these reasons to dispel patient fears or concerns and motivate them to use the brush.

In this study, the BOFIB index was used to assess the effects of interdental brushing. Previous studies have evaluated changes in bleeding using indices such as EIB, BOIB, or BOB; however, these indices only evaluate bleeding and may not reflect the interproximal plaque. Plaque is the primary factor in periodontal inflammation, and effective plaque removal is the basis for periodontal management. The BOFIB index allows quick assessment of interdental health and may help identify the interproximal plaque for removal.

In this oral health promotion program, the participants were educated about the use of interdental brushing. The information leaflet included predictable complications, such as pain, bleeding, or gingival recession, which could be acceptable during the healing of gingivitis. Unfortunately, 17 of the 26 participants in the IB-NN group had not used the interdental brush more than twice per week during the six months of the program. Moreover, 11 participants in the IB-NN group reported that interdental brushing was uncomfortable or unnecessary. These findings show the difficulties encountered with behavioral change in oral health education.

This study had certain limitations. First, the sample size was relatively small because of the long-term follow-up (three and six months). Moreover, most people who do not experience oral pain (e.g., patients with mild periodontitis) seldom visit dental clinics or hospitals. For these reasons, only 26 of the 76 participants from the first visit remained in the study at six months. Second, the periodontal assessment was performed using BOFIB and not clinical attachment loss (CAL). Therefore, we did not assess the periodontal attachment level because the results were obtained from an oral health promotion program. However, this study used retrospective oral health promotion data, and participants who needed periodontal treatment were excluded because they should receive periodontal treatment in dental clinics or hospitals. Obtaining CAL and pocket depth measurements of the sextant area of the mouth is time consuming, especially in a busy clinic. Moreover, if recorded, these values may have a large standard deviation [18,19], making them unreliable without deliberate calibration training.

Regular visits to dental clinic in the current COVID-19 era are becoming more difficult, which can exacerbate the deterioration of oral health. The management of periodontal patients can be permanently adjusted, including self-management through a variety of oral care tools, less complex treatment and a more comprehensive and deterministic approach [20]. The results of this study showed that BOFIB was relatively well maintained by those who were educated to manage their teeth at least once every two days [21] and who continuously managed for three or more times per week. Therefore, interdental brush can be recommended using it at least once every two days in consideration of individual oral health status and oral health related habits such as smoking. Interdental brushes, which are representative tools for self-oral health management of periodontal disease, can use the state of periodontal health as a self-assessment and an evaluable index by experts, as in this study. It is considered that it can be used as a research result that can promote the global oral health in the non-face-to-face era by expanding and disseminating through subsequent research and presenting accurate indicators for self-oral health management.

Nevertheless, we evaluated the effects of interdental brushing on interproximal plaque according to patient compliance at a baseline, 3-month, and 6-month follow-up. Such a study has rarely been conducted. When planning future large-scale cross-sectional studies or longitude follow-up studies that measure the BOFIB comparing CALs and BOPs, it is possible that this study’s results will be served as a pilot study when calculating sample counts. However, our data may present a basis for patient education and motivate them to use an interdental brush, and the results should be discussed and how they can be interpreted from the perspective of previous studies and of the working hypotheses. The findings and their implications should be discussed in the broadest context possible. Future research directions may also be highlighted.

## 5. Conclusions

In conclusion, the BOFIB index score of participants who were compliant with the triweekly interdental brushing education was lower than that of the noncompliant participants after using it for 3 and 6 months.

## Figures and Tables

**Figure 1 healthcare-09-01239-f001:**
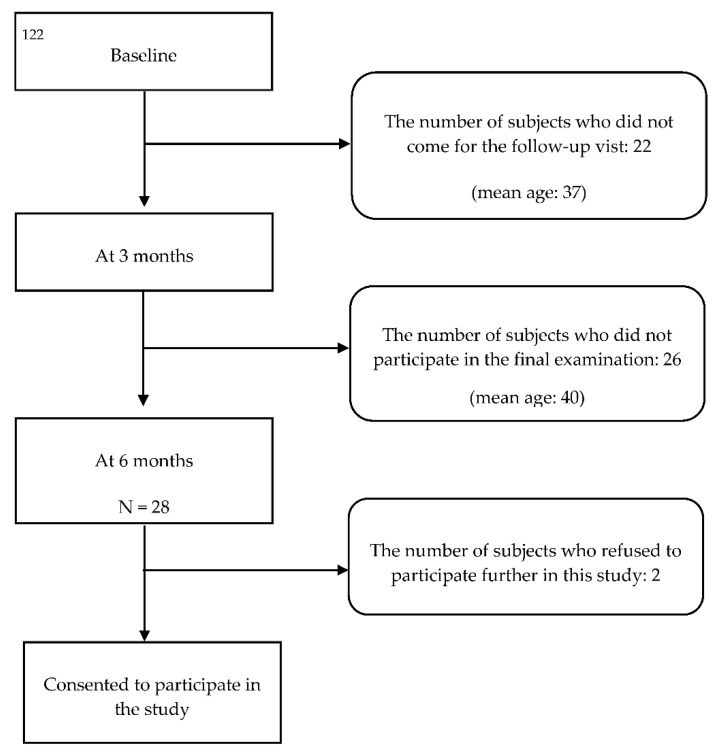
Flow chart describing the number of participants at each study stage.

**Table 1 healthcare-09-01239-t001:** Information about the size of the interdental brush.

Wire Diameter	Brush Size (mm)	Minimum AccessDiameter	Subjects (%)
0.23 mm	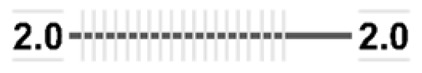	0.7 mm	3 (11.1%)
0.25 mm	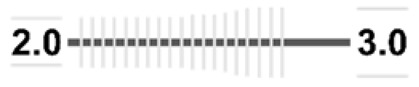	0.8 mm	1 (3.7%)
0.27 mm	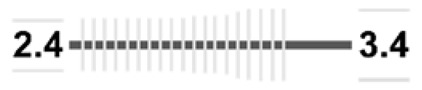	1.0 mm	21 (77.8%)
0.30 mm	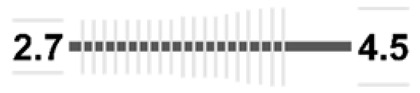	1.2 mm	2 (7.4%)

**Table 2 healthcare-09-01239-t002:** Bleeding on the full-mouth interdental brush (BOFIB) index.

Index	Figure	Explanation
BOFIB0	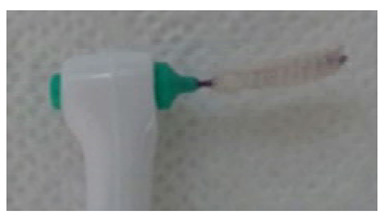	No bleeding and only little white plaque noted after full-mouth interdental brushing
BOFIB1	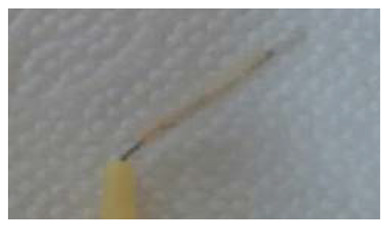	No bleeding and moderate yellowish plaque noted after full-mouth interdental brushing
BOFIB2	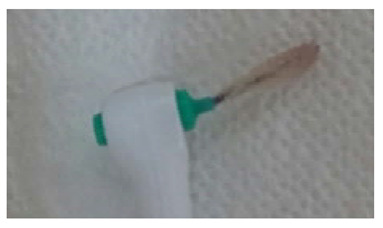	Little bleeding and little red color noted after full-mouth interdental brushing
BOFIB3	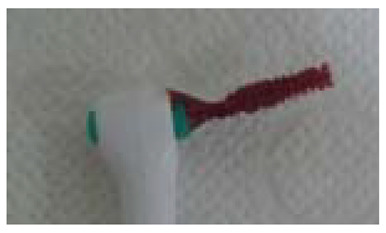	Abundant bleeding after full-mouth interdental brushing
BOFIB4	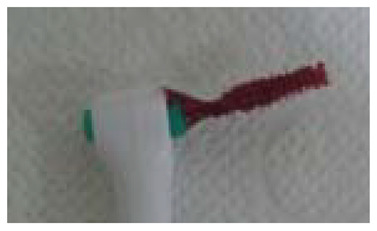	Abundant bleeding during full-mouth interdental brushing

**Table 3 healthcare-09-01239-t003:** Characteristics of the study population according to interdental brushing compliance (*N* = 26).

	IB-NN Group (*n* = 17)	IB-NY Group (*n* = 5)	IB-YY Group (*n* = 4)	*p* Value *
Age	36.41 ± 10.57 ^a^	51.80 ± 5.89 ^b^	52.00 ± 2.45 ^a,b^	0.009
Gender				
Men	10 (58.8%)	5 (100%)	2 (50%)	0.174
Women	7 (41.2%)	0 (0%)	2 (50%)	
Diabetes Mellitus				
No	17 (100%)	4 (80.0%)	3 (75.0%)	0.111
Yes	0 (0%)	1 (20.0%)	1 (25.0%)	
Smoking				
No	11 (64.7%)	1 (20.0%)	4 (100%)	0.148
Past	4 (11.8%)	2 (40.0%)	0 (0%)	
Current	2 (23.5)	2 (40.0%)	0 (0%)	
Major interdental brush size				
0.23 mm	3 (17.6%)	0 (0%)	0 (0%)	0.359
0.25 mm	1 (5.9%)	0 (0%)	0 (0%)	
0.27 mm	13 (76.5%)	5 (100%)	3 (75.0%)	
0.30 mm	0 (0%)	0 (0%)	1 (25.0%)	
Weekly number of interdental flossing				
	0.97 ± 1.77	0.50 ± 0.71	0.38 ± 0.75	0.430
Weekly number of interdental brushing at baseline				
	0.15 ± 0.34 ^a^	0.80 ± 1.30 ^a^	10.00 ± 7.39 ^b^	0.001
Weekly number of interdental brushing at 6 months				
	0.53 ± 0.86 ^a^	6.60 ± 0.89 ^b^	10.50 ± 7.00 ^b^	<0.001

Abbreviations: IB-NN group, subjects used the interdental brush less than three times per week both before and after the program; IB-NY group, subjects used the interdental brush less than three times per week before the program but used it at least three times per week after the program; IB-YY group, subjects used the interdental brush at least three times per week both before and after the program. * Analyzed by an independent samples t-test for age and weekly number of interdental flossing and chi-square test for other variables. ^a,b^ Different character means a significant difference between the groups based on the results of Mann–Whitney U post hoc analysis.

**Table 4 healthcare-09-01239-t004:** Bleeding on full-mouth interdental brushing (BOFIB) index according to full-mouth interdental brushing (*N* = 26).

	IB-NN Group ^a^ (*n* = 17)	IB-NY Group ^a,b^ (*n* = 5)	IB-YY Group ^b^ (*n* = 4)	*p* value *
Baseline	2.88 ± 0.80	3.00 ± 1.00	1.75 ± 1.26	<0.001
3 months	2.65 ± 0.63	2.30 ± 0.97	1.13 ± 0.63	
6 months	2.44 ± 0.58	1.40 ± 1.19	1.00 ± 1.82	

Abbreviations: IB-NN group, subjects used the interdental brush less than three times per week both before and after the program; IB-NY group, subjects used the interdental brush less than three times per week before the program but used it at least three times per week after the program; IB-YY group, subjects used the interdental brush at least three times per week both before and after the program. * Analyzed by repeated measures ANOVA. ^a,b^ Different characters mean significant difference between groups in Scheffe’s post hoc analysis.

**Table 5 healthcare-09-01239-t005:** Reasons for less use of interdental brushing in the IB-NN group (*n* = 17).

Reason	Number (Redundant Answer)
Had black triangle concern (*n* = 17)	1 (5.9%)
Experienced pain (*n* = 17)	2 (11.8%)
Uncomfortable and non-habitual (*n* = 17)	11 (64.7%)
Did not purchase (*n* = 17)	2 (11.8%)
Seemed unnecessary (*n* = 17)	5 (29.4%)

## Data Availability

The datasets used or analyzed during the current study are available from the corresponding author on reasonable request.

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
