# Peer review of "Effect of Triweekly Interdental Brushing on Bleeding Reduction in Adults: A Six-Month Retrospective Study"

_healthcare, 2021, doi:10.3390/healthcare9091239_

Round 1
Reviewer 1 Report
The topic of the present six-month retrospective study, evaluating the effect of triweekly interdental brushing on bleeding reduction, is interesting and the findings may be clinically relevant to provide specific recommendations for the interdental brushing (timing and technique), based on periodontal status (periodontal health, gingivitis, periodontitis stages). However, Methods should be largely improved.
Concerns and Suggestions:
Introduction:
- Please, re-phrase the periods in lines 68-76 in order to clarify the aim(s) of the investigation.
Materials and Methods:
- Please, add sample size calculation (26 subjects?), inclusion/exclusion criteria and follow-up schedule (paragraph 2.3 Procedure).
Results:
- Please, add informations regarding partecipants' periodontal status (specifying that CAL and other periodontal paramenters have not been assessed, as reported in lines 285-287), which may be particularly important in providing specific recommendations for interdental brushing (timing and technique) based on periodontal status (periodontal health, gingivitis, periodontitis stages).
Discussion:
- Please add the related references to “previous studies” (line 226).
- Please, re-phrase the period in lines 293-298.
Author Response
Thank you for your constructive and excellent comments which have helped us to improve our manuscript, “Effect of triweekly interdental brushing on bleeding reduction in adults: a six-month retrospective study” (healthcare-1354152). We revised our manuscript according to the comments from the reviewers and the revised parts were highlighted in red color in the revised manuscript. Important changes were also described in this response letter point-by-point. I hope that you will find these alterations satisfactory.
We would like to thank the editors and reviewers for their suggestions and look forward to having our manuscript published in Healthcare.
Sincerely,
*Corresponding author:
Hyun-Jae Cho
Department of Preventive Dentistry & Public Oral Health, School of Dentistry, Seoul National University, 101 Daehakro, Jongno-gu, Seoul, Republic of Korea 03080
Tel: +82 (0)2 740-8677
Fax: +82 (0)2 765-1722
E-mail: [email protected]
*Corresponding author:
Hyo-Jin Lee
Department of Dental Hygiene, College of Dentistry and Research Institute of Oral Science, Gangneung-Wonju National University, Gangeung, Republic of Korea 25457
Tel: +82 (0)33 640-3028
E-mail: [email protected]
Response to comments from the Reviewer 1:
- Introduction:
Please, re-phrase the periods in lines 68-76 in order to clarify the aim(s) of the investigation.
We reviewed the relevant aim parts in the introduction and revised it as follows:
“The purpose of this study is to identify the bleeding reduction effect of the triweekly interdental toothbrush using the bleeding on full-mouth interdental brushing (BOFIB) index, over a period of 6 months.”
- Materials and Methods:
Please, add sample size calculation (26 subjects?), inclusion/exclusion criteria and follow-up schedule (paragraph 2.3 Procedure).
Thank you for your comments. We added the sample size estimation as follows:
“The appropriate sample size for the study was calculated using a power and sample size calculation program (G Power 3.1.9.7) to estimate a sample size sufficient to detect differences in the BOFIB index between the groups based on interdental brushing compliance. The type I error probability (α) was 0.05, and the power (1-β) was 0.8. We initially estimated a sample of 26 participants but decided on a total sample size of 75 participants to account for possible loss to follow-up.”
- Results:
Please, add informations regarding partecipants' periodontal status (specifying that CAL and other periodontal paramenters have not been assessed, as reported in lines 285-287), which may be particularly important in providing specific recommendations for interdental brushing (timing and technique) based on periodontal status (periodontal health, gingivitis, periodontitis stages).
The collected data was obtained as part of a health promotion program, so unfortunately no information such as CAL was collected. This point is additionally described in the limitation.
- Discussion:
Please add the related references to “previous studies” (line 226).
Thank you for your indication. We added references.
- Please, re-phrase the period in lines 293-298.
We revised the relevant sentence as follows:
“Nevertheless, we evaluated the effects of interdental brushing on interproximal plaque according to patient compliance at baseline, a 3-month, and a 6-month follow-up.”
Reviewer 2 Report
Dear Authors, following are some minor and major suggestions.
Minor:
Please make the font of the text uniform.
Standardize the text in table 2, in particular to the line "BOFIB3" and in table 3, for example, the word “gender”, or the line of the "men".
Correct some typos as in paragraph 2.6, line 166, where the groups are listed (IB-NN, IB-NY, and IB-NY. IB-YY is omitted) or in line 295 "brush.rs" or in line 161 “porticipants”.
References: check number 8 standardizing the format as required by the guideline.
Majors:
In material and methods, I suggest revising table 1 reporting the number at the baseline of the population, in order to have a prompt indication of the subjects for each clinical stage
In materials e methods, paragraph 2.2, must be better defined semiquantitative evaluation system used in the present study, about blood and plaque evaluation remained after interdental area cleaning during dentist clinical assessment.
In the Discussion, the Authors should define if this study is a pilot/preliminary study propaedeutical to other clinical future research projects effectuated on a major number of participants in order to reinforce statistical evidence. Furthermore, in my opinion, the article needs to be more contextualized highlighting the importance of the obtained results on the population.
Author Response
Thank you for your constructive and excellent comments which have helped us to improve our manuscript, “Effect of triweekly interdental brushing on bleeding reduction in adults: a six-month retrospective study” (healthcare-1354152). We revised our manuscript according to the comments from the reviewers and the revised parts were highlighted in red color in the revised manuscript. Important changes were also described in this response letter point-by-point. I hope that you will find these alterations satisfactory.
We would like to thank the editors and reviewers for their suggestions and look forward to having our manuscript published in Healthcare.
Sincerely,
*Corresponding author:
Hyun-Jae Cho
Department of Preventive Dentistry & Public Oral Health, School of Dentistry, Seoul National University, 101 Daehakro, Jongno-gu, Seoul, Republic of Korea 03080
Tel: +82 (0)2 740-8677
Fax: +82 (0)2 765-1722
E-mail: [email protected]
*Corresponding author:
Hyo-Jin Lee
Department of Dental Hygiene, College of Dentistry and Research Institute of Oral Science, Gangneung-Wonju National University, Gangeung, Republic of Korea 25457
Tel: +82 (0)33 640-3028
E-mail: [email protected]
Response to comments from the Reviewer 2:
- Please make the font of the text uniform.
We unified the manuscript's fonts to “Palatino Linotype”.
- Standardize the text in table 2, in particular to the line "BOFIB3" and in table 3, for example, the word “gender”, or the line of the "men".
Thank you for your comments. We rearranged the lines of the text in the table.
- Correct some typos as in paragraph 2.6, line 166, where the groups are listed (IB-NN, IB-NY, and IB-NY. IB-YY is omitted) or in line 295 "brush.rs" or in line 161 “porticipants”.
Thank you for your indication. We identified the typos and revised them.
- References: check number 8 standardizing the format as required by the guideline.
We revised the reference format.
- In material and methods, I suggest revising table 1 reporting the number at the baseline of the population, in order to have a prompt indication of the subjects for each clinical stage
The size of the interdental brush is measured only in the baseline and there is no change, so we added columns to Table 1 to indicate the number of subjects by size not by each clinical stage.
- In materials e methods, paragraph 2.2, must be better defined semiquantitative evaluation system used in the present study, about blood and plaque evaluation remained after interdental area cleaning during dentist clinical assessment.
Thank you for your indication. The process of obtaining BOFIB is described in a more systematic and sequential way as below.
“2.2.1 Selection of an interdental brush size” and “2.2.2 Cleansing of interdental areas using the interdental brush” in materials and methods parts.
- In the Discussion, the Authors should define if this study is a pilot/preliminary study propaedeutical to other clinical future research projects effectuated on a major number of participants in order to reinforce statistical evidence. Furthermore, in my opinion, the article needs to be more contextualized highlighting the importance of the obtained results on the population.
Thank for your kind suggestion. We added the additional sentence in the end the discussion section as below.
“Nevertheless, we evaluated the effects of interdental brushing on interproximal plaque according to patient compliance after a 6-month follow-up. Such a study has rarely been conducted. When planning future large-scale cross-sectional studies or longitude follow-up studies that measure the BOFIB comparing CALs and BOPs, our study result will be served as pilot study when calculating sample counts. On the other side, our data could present a basis for patient education and motivate them to use an interdental brushes should discuss the results and how they can be interpreted from the perspective of previous studies and of the working hypotheses. The findings and their implications should be discussed in the broadest context possible. Future re-search directions may also be highlighted.”
Round 2
Reviewer 1 Report
Some further concerns and suggestions:
Materials and Methods:
- Please, use only the past tense “Based on Table 1, a dentist (H-J) with extensive experience in the use of an interdental 89 toothbrush visually examined the interdental space of the subjects and determines the size 90 of the interdental brush (from SSS to M table 1) that can pass through almost all of the 91 interdental areas of the subjects (> 90%)” (lines 89-91);
- Table 1: Please, remove the columns “Size” and “Color”;
- Please, unify sub-paragraph 2.2.2 to the 2.2.3 not modifying the title “2.2.3 Decision of the BOFIB index”.
Discussion:
- Please, correct “receive” (line 301)
- Please, implement the period in lines 306-311, concerning periodontal care, and particularly self-care, in the context of the COVID-19, see an inherent very recent article “Evidence-based Recommendations on Periodontal Practice and the Management of Periodontal Patients during and after the COVID-19 Era: Challenging Infectious Diseases Spread by Airborne Transmission”; it might be clinically relevant unify periodontal self-care recommendations, strategies and suggestions;
- “Results” and not “result” (line 317), and re-write the period in lines 315-317 to make it more readable.
Author Response
First of all, I would like to thank you for giving me a detailed review opinion to improve the quality of the paper.
Materials and Methods:
R1) Please, use only the past tense “Based on Table 1, a dentist (H-J) with extensive experience in the use of an interdental 89 toothbrush visually examined the interdental space of the subjects and determines the size 90 of the interdental brush (from SSS to M table 1) that can pass through almost all of the 91 interdental areas of the subjects (> 90%)” (lines 89-91);
It has been modified to reflect the review opinions. (line 89-91)
R2) Table 1: Please, remove the columns “Size” and “Color”;
It has been modified to reflect the review opinions. (Table 1, Table 2)
R3) Please, unify sub-paragraph 2.2.2 to the 2.2.3 not modifying the title “2.2.3 Decision of the BOFIB index”.
It has been modified to reflect the review opinions. (line 96-99)
Discussion:
R4) Please, correct “receive” (line 301)
It has been modified to reflect the review opinions. (line 299)
R5) Please, implement the period in lines 306-311, concerning periodontal care, and particularly self-care, in the context of the COVID-19, see an inherent very recent article “Evidence-based Recommendations on Periodontal Practice and the Management of Periodontal Patients during and after the COVID-19 Era: Challenging Infectious Diseases Spread by Airborne Transmission”; it might be clinically relevant unify periodontal self-care recommendations, strategies and suggestions;
By reflecting the review opinion, the appropriate management period according to the results of the study was suggested, and references were added to reinforce it. (line 304-312)
R6) “Results” and not “result” (line 317), and re-write the period in lines 315-317 to make it more readable.
It has been modified to reflect the review opinions. (line 321-323)
Reviewer 2 Report
Dear Authors,
thank you for the applied revision. The article now results improve and in line with all the suggestions.
Best Regards
Author Response
I would like to thank you for giving me a detailed review opinion to improve the quality of the paper.